# Impact of Physical Activity on Overall Survival and Liver Cirrhosis Incidence in Steatotic Liver Disease: Insights from a Large Cohort Study Using Inverse Probability of Treatment Weighting

**DOI:** 10.3390/nu16152532

**Published:** 2024-08-02

**Authors:** Keungmo Yang, Beom Sun Chung, Tom Ryu

**Affiliations:** 1Department of Internal Medicine, Division of Gastroenterology and Hepatology, College of Medicine, The Catholic University of Korea, Seoul 06591, Republic of Korea; yang27jin@catholic.ac.kr; 2Department of Anatomy, College of Medicine, Yonsei University Wonju, Wonju 26426, Republic of Korea; 3Department of Internal Medicine, Institute for Digestive Research, Digestive Disease Center, College of Medicine, Soonchunhyang University, Seoul 04401, Republic of Korea

**Keywords:** exercise, liver cirrhosis, overall survival, inverse probability treatment weighting

## Abstract

Physical activity is a cornerstone of a healthy lifestyle, with benefits in managing chronic diseases. This study investigates the relationship between physical activity and liver-related outcomes with or without steatotic liver diseases, including metabolic dysfunction-associated steatotic liver disease (MASLD) and MASLD and increased alcohol intake (MetALD). The primary outcomes of interest were overall survival in the entire population, individuals without steatotic liver disease, patients with MASLD, and those with MetALD. The secondary outcomes included the incidence of liver cirrhosis. Participants were categorized based on physical activity frequency and Kaplan–Meier survival curves and Cox proportional hazards models were used for analysis. Higher physical activity was associated with significantly better survival in the overall cohort and MASLD cohort before and after inverse probability of treatment weighting (IPTW). In participants without steatotic liver disease and the MetALD cohort, higher physical activity showed significant survival improvement after IPTW. For the incidence of liver cirrhosis, higher physical activity showed significant associations before IPTW in the overall cohort and MASLD cohort, but these associations were not significant after IPTW. Marginal significance was observed in the MetALD cohort before and after IPTW. In conclusion. promoting physical activity may be key in improving liver-related outcomes.

## 1. Introduction

Physical activity has long been recognized as a cornerstone of a healthy lifestyle, playing a critical role in the prevention and management of a wide range of chronic diseases. The benefits of regular exercise are well-documented, extending beyond cardiovascular health to include metabolic and mental well-being [1,2,3]. Among its numerous advantages, physical activity has been shown to reduce the risk of obesity, type 2 diabetes, hypertension, and dyslipidemia, all of which are key risk factors for non-communicable diseases [4,5,6].

Liver diseases, particularly metabolic dysfunction-associated steatotic liver disease (MASLD) and its mixed nomenclature with alcohol consumption, known as MASLD and increased alcohol intake (MetALD), are emerging as major public health challenges worldwide [7]. MASLD, the term which represents fatty liver disease nowadays, is characterized by the accumulation of fat in the liver in the absence of significant alcohol consumption, and it is strongly associated with metabolic syndrome [8,9]. The global prevalence of MASLD is estimated to be about 30%, with a 50% increase for 20 years [10]. The disease spectrum ranges from MASLD to more severe conditions such as fibrosis, cirrhosis, and hepatocellular carcinoma, highlighting the critical need for effective prevention and management strategies [11].

There is growing evidence to suggest that physical activity exerts a protective effect against liver diseases. Exercise is believed to improve hepatic lipid metabolism, reduce insulin resistance, and decrease systemic inflammation, all of which are critical mechanisms in the pathogenesis of MASLD [12]. Studies have shown that both aerobic and resistance training can lead to reductions in liver fat content and improvements in liver enzyme levels [13,14]. For instance, aerobic exercise has been associated with significant reductions in intrahepatic fat and improvements in insulin sensitivity, while resistance training has been shown to decrease liver fat and improve muscle mass and strength [15,16,17]. Despite these promising findings, the optimal type, intensity, and frequency of exercise required to confer these benefits remain areas of active research. Understanding these parameters is crucial for developing targeted interventions that maximize the protective effects of physical activity on liver health.

Meanwhile, the UK Biobank, a large-scale biomedical database and research resource, offers a unique opportunity to explore the associations between lifestyle factors, such as physical activity, and health outcomes in a diverse cohort of over 500,000 participants. This extensive resource includes detailed information on participants’ genetic, physical, and health-related characteristics, providing a rich dataset for investigating long-term health effects and identifying modifiable risk factors for various diseases, including liver conditions. The UK Biobank’s comprehensive data collection enables researchers to control for a wide range of confounding variables, thereby enhancing the reliability and validity of findings.

Here, we investigated the relationship between physical activity frequency, incidence of liver cirrhosis, and overall survival in a large diverse cohort with or without steatotic liver diseases. To ensure robust and unbiased comparisons between groups with different physical activity frequencies, this study employed propensity score matching using inverse probability of treatment weighting (IPTW). This statistical technique balances the cohorts on various baseline characteristics, minimizing confounding factors that could influence the outcomes [18]. By applying IPTW, this study aims to create comparable groups that differ primarily in their physical activity levels, thus allowing for a clearer interpretation of the effect of physical activity on liver disease incidence and overall survival.

## 2. Materials and Methods

### 2.1. Study Population Collected

This retrospective cohort study utilized data from the UK Biobank (Application ID: 117214), a large-scale biomedical database that includes detailed health and lifestyle information from over 500,000 participants across the United Kingdom. Participants were recruited between 2006 and 2010 and provided comprehensive baseline data through questionnaires, physical measurements, and biological samples.

This study was approved by the relevant ethics committees, and all participants provided informed consent. The UK Biobank has obtained ethical approval from the North West Multi-Centre Research Ethics Committee (MREC) and is compliant with the principles of the Declaration of Helsinki. The research was conducted in accordance with the guidelines and regulations set forth by the UK Biobank.

### 2.2. Data Collection and Serological Markers

Baseline data included demographic information (age, sex), lifestyle factors (physical activity frequency, smoking status), and clinical characteristics (body mass index [BMI], waist circumference, history of type 2 diabetes, dyslipidemia, and hypertension). Liver function tests were also collected, including alanine aminotransferase (ALT), gamma-glutamyl transferase (GGT), platelet count, and albumin levels.

Alcohol intake data were collected through self-reported questionnaires and included in the analysis to differentiate between MASLD and MetALD. For statistical analysis, alcohol consumption was categorized according to established thresholds (20–50 g/day for females and 30–60 g/day for males) to identify MetALD patients. This approach follows established guidelines for defining and differentiating MASLD and MetALD.

### 2.3. Physical Activity Assessment

Physical activity frequency was self-reported by participants and categorized into two groups: those engaging in physical activity ≥ 4 days per week and those engaging <3 days per week. This classification was based on responses to questions regarding the frequency and intensity of physical activities, including walking, moderate activity, and vigorous activity.

### 2.4. Primary Outcome and the Definition of Steatotic Liver Disease

Liver disease outcomes were identified using hospital records, death registries, and self-reported data. The primary outcomes of interest were the overall survival of the entire population, people without steatotic liver disease, and patients with MASLD and MetALD. The secondary outcomes were the incidence of liver cirrhosis with the cohort classified above. MASLD was defined as the presence of fatty liver in the absence of significant alcohol consumption, consistent with established diagnostic cardiometatbolic criteria, and MetALD was defined as MASLD with a daily intake of 20 to 50 g of alcohol (or weekly 140–350 g) for females and 30 to 60 g daily for males (or weekly 210–420 g) [19].

### 2.5. Application of IPTW

To account for potential confounding variables and ensure balanced comparisons between the physical activity groups, we employed propensity score matching using inverse probability of IPTW.

First, the propensity score for each participant was estimated using a logistic regression model. This model included a range of covariates known to be associated with both physical activity and liver-related outcomes. The covariates included in the propensity score model were demographic variables (age, sex), lifestyle factors (smoking status), clinical characteristics (BMI, waist circumference, history of type 2 diabetes, dyslipidemia, hypertension), and liver function tests (ALT, GGT, platelet count, albumin levels). These covariates were chosen based on their established associations with liver disease and overall survival, ensuring that the model accounts for major confounders that could influence the outcomes.

Next, inverse probability weights were calculated as the inverse of the propensity score for participants in the higher physical activity group (≥4 days/week) and as the inverse of 1 minus the propensity score for participants in the lower physical activity group (<3 days/week). To ensure that IPTW achieved balance between the groups, standardized mean differences (SMDs) were calculated for all covariates. An SMD of less than 0.1 was considered indicative of adequate balance.

### 2.6. Statistical Analysis

To account for potential confounding variables and ensure balanced comparisons between the physical activity groups, propensity score matching using IPTW was employed. The propensity score for each participant was estimated using a logistic regression model, which included age, sex, BMI, waist circumference, smoking status, history of type 2 diabetes, dyslipidemia, hypertension, ALT, GGT, platelet count, and albumin levels.

Descriptive statistics were used to summarize the baseline characteristics of the study population before and after IPTW. Continuous variables were presented as mean ± standard deviation (SD), and categorical variables were expressed as frequencies and percentages. Standardized mean differences (SMD) were calculated to assess the balance of covariates between the groups, with an SMD of <0.1 indicating adequate balance.

Cox proportional hazards models were used to estimate hazard ratios (HRs) and 95% confidence interval (CI) for the association between less physical activity frequency (<3 days per week) and the overall survival. Analyses were conducted separately for the entire cohort and stratified by the presence of steatotic liver diseases such as MASLD and MetALD. Models were adjusted for potential confounders, including demographic, lifestyle, and clinical characteristics.

We constructed a Kaplan–Meier curve of overall survival for all-cause mortality and the incidence of liver cirrhosis for the entire cohort, those without steatotic liver disease, and the MASLD and MetALD cohort, with a *p*-value of significance. The log-rank test was employed to assess the statistical significance of survival differences between groups. The observation period for this study spanned up to 18 years, with an average follow-up period of approximately 15 years.

Additionally, alcohol intake was included as a covariate in the logistic regression models used to calculate propensity scores for IPTW. By adjusting for alcohol consumption, we aimed to control for its confounding effect on liver disease outcomes. Specifically, participants were categorized into different groups based on their reported alcohol intake, and these categories were used to adjust the propensity scores. This allowed us to isolate the effect of physical activity on liver disease outcomes while accounting for variations in alcohol consumption.

### 2.7. Statistical Software

All statistical analyses were performed using R (version 4.4.0; R Foundation for Statistical Computing, Vienna, Austria). Propensity score matching and IPTW were conducted using the “twang” package, and Cox proportional hazards models were fitted using the “survival” package. A two-sided *p*-value of <0.05 was considered statistically significant.

## 3. Results

### 3.1. Baseline Characteristics of the Cohort

The baseline clinical characteristics of the study population are summarized in Table 1, Table 2, Table 3 and Table 4. The cohort was divided based on physical activity frequency: those engaging in physical activity ≥ 4 days per week and those with <3 days per week. Propensity score matching using IPTW was applied to balance the baseline characteristics between these groups.

Before IPTW, significant differences were observed between the groups in terms of sex, age, body mass index (BMI), waist circumference, and prevalence of type 2 diabetes, dyslipidemia, and hypertension in entire cohort. After IPTW, these differences were minimized, indicating successful balancing of the groups. For example, the standardized mean difference (SMD) for age reduced from 0.116 to 0.011, and for BMI from 0.213 to 0.002 (Table 1).

In the cohort without steatotic liver disease, similar trends were observed. Before IPTW, significant differences existed in sex distribution, age, and waist circumference. Post-IPTW, these differences were reduced, with SMDs for age and BMI being 0.007 and 0.003, respectively (Table 2).

For participants with MASLD, baseline differences in sex, age, BMI, and waist circumference were noted before IPTW. After IPTW, these characteristics were balanced, with SMDs for age and BMI being 0.010 and 0.001, respectively (Table 3).

In the cohort with MetALD, baseline characteristics such as sex, age, BMI, and waist circumference showed significant differences before IPTW, which were balanced post-IPTW, with SMDs for age and BMI being 0.012 and 0.003, respectively (Table 4).

### 3.2. Impact of Physical Activity on All-Cause Mortality

The impact of physical activity on all-cause mortality was analyzed using the Cox proportional hazards models. Both univariate and multivariate analyses were conducted before and after IPTW.

Before IPTW, lower physical activity (<3 days per week) was associated with a significantly higher risk of all-cause mortality (HR 1.12, 95% CI 1.10–1.15, *p* < 0.001). After IPTW, this association remained significant (HR 1.12, 95% CI 1.10–1.15, *p* < 0.001).

In participants without steatotic liver disease, lower physical activity was not significantly associated with all-cause mortality before IPTW (HR 1.00, 95% CI 0.95–1.06, *p* = 0.890). However, after IPTW, a significant association was observed (HR 1.16, 95% CI 1.10–1.23, *p* < 0.001).

For the MASLD cohort, lower physical activity was associated with a higher risk of all-cause mortality both before (HR 1.13, 95% CI 1.09–1.16, *p* < 0.001) and after IPTW (HR 1.12, 95% CI 1.09–1.16, *p* < 0.001).

In the MetALD cohort, lower physical activity was not significantly associated with all-cause mortality before IPTW (HR 1.02, 95% CI 0.97–1.02, *p* = 0.420). After IPTW, a marginally significant association was observed (HR 1.07, 95% CI 1.00–1.14, *p* = 0.039) (Table 5).

### 3.3. Overall Survival for All-Cause Mortality Using Kaplan–Meier Analysis

Kaplan–Meier survival curves were used to visualize the overall survival (OS) probability over time for different cohorts categorized by physical activity frequency. Log-rank tests were conducted to compare survival distributions.

The Kaplan–Meier curves indicated that higher physical activity (≥4 days per week) was associated with significantly better OS compared to lower physical activity (<3 days per week) before IPTW (*p* < 0.0001). After IPTW, the difference in OS remained significant (*p* < 0.0001) (Figure 1A and Figure 2A). In the cohort without steatotic liver disease, higher physical activity was not significantly associated with better OS before IPTW (*p* = 0.890). Nevertheless, post-IPTW, the difference showed significance (*p* < 0.0001) (Figure 1B and Figure 2B). For participants with MASLD, higher physical activity was associated with significantly better OS before IPTW (*p* < 0.0001) and after IPTW (*p* < 0.0001) (Figure 1C and Figure 2C). In the MetALD cohort, higher physical activity was not associated with better OS before IPTW (*p* = 0.420), while after IPTW, the association was presented to be marginally significant (*p* = 0.044) (Figure 1D and Figure 2D). Collectively, higher physical activity (≥4 days per week) was consistently associated with significantly better overall survival (OS) in the overall cohort and in participants with MASLD before and after IPTW. Moreover, although participants without steatotic liver disease and MetALD did not appear to have significant alterations in OS between the two groups of exercise, post-IPTW data showed that higher physical activity has significance for better OS in the group of higher physical activity compared to lower physical activity in the two cohort.

### 3.4. Incidence of Liver Cirrhosis of Various Steatotic Liver Disease with Physical Activity

Kaplan–Meier curves for the incidence of liver cirrhosis showed that participants with higher physical activity had marginally lower incidence rates compared to those with lower physical activity after IPTW. Higher physical activity (≥4 days per week) was associated with a significantly lower incidence of liver cirrhosis compared to lower physical activity (<3 days per week) before IPTW (*p* < 0.0001), while this significant association did not remain after IPTW (*p* = 0.278) (Figure 3A and Figure 4A). In the cohort without steatotic liver disease, before IPTW, higher physical activity was not associated with lower liver cirrhosis incidence (*p* = 0.250), and the insignificance remained after IPTW (*p*= 0.444) (Figure 3B and Figure 4B). Higher physical activity was significantly associated with a lower incidence of liver cirrhosis before IPTW (*p* < 0.0001), and the significance disappeared with adjustment with IPTW in MASLD cohort (*p* = 0.777) (Figure 3C and Figure 4C). Finally, higher physical activity was associated with a marginal significance of a lower incidence of liver cirrhosis before (*p* = 0.032) and after IPTW (*p* = 0.048) (Figure 3D and Figure 4D). Overall, after IPTW adjustment, higher physical activity was not significantly associated with a lower incidence of liver cirrhosis in the overall cohort, the cohort without steatotic liver disease, and the MASLD cohort. Only in the MetALD cohort was a marginally significant association observed both before and after IPTW adjustment.

## 4. Discussion

The findings of this study highlight the significant role of physical activity in improving overall survival (OS) and its marginal effect on reducing the incidence of liver cirrhosis, particularly in individuals with MASLD and MetALD. MASLD was defined as the presence of hepatic steatosis in the absence of significant alcohol intake, according to established criteria. MetALD was defined as MASLD with a daily alcohol consumption of 20–50 g for females and 30–60 g for males [20]. Liver cirrhosis was diagnosed based on clinical, laboratory, and imaging findings consistent with advanced liver fibrosis and impaired liver function. The criteria for diagnosing cirrhosis included a combination of at least two of the following: liver biopsy showing Ishak stage 5–6, imaging evidence of a nodular liver with signs of portal hypertension, liver stiffness measurement exceeding specific thresholds, presence of varices, and blood-based biomarkers indicating advanced fibrosis [21].

The evident association between higher physical activity (≥4 days per week) and improved OS observed in this study is consistent with the existing literature that emphasizes the multifaceted benefits of regular exercise. Physical activity is known to enhance cardiovascular health, improve metabolic functions, and bolster mental well-being, thereby contributing to overall longevity [1]. The findings that this association holds true even after adjusting for a wide range of confounding variables using inverse probability of treatment weighting (IPTW) further solidify the evidence that physical activity is a crucial determinant of survival.

In the overall cohort, participants who engaged in physical activity ≥ 4 days per week exhibited significantly better OS compared to those with <3 days per week of physical activity. This relationship persisted both before and after IPTW adjustment, demonstrating the independent effect of physical activity on survival outcomes. Similarly, in participants with MASLD, higher physical activity was consistently associated with better OS before and after IPTW. These findings suggest that regular physical activity may mitigate the adverse effects of MASLD—a condition characterized by hepatic fat accumulation and associated metabolic dysfunctions.

Interestingly, in the cohort without steatotic liver disease, higher physical activity did not significantly correlate with better OS before IPTW but showed significant improvement post-IPTW. This discrepancy indicates that confounding factors might have obscured the benefits of physical activity in the unadjusted analysis. The IPTW adjustment likely controlled for these confounders, revealing the true beneficial impact of physical activity.

In the MetALD cohort, the association between physical activity and OS was not significant before IPTW, while it became marginally significant after IPTW adjustment. This marginal significance suggests that while physical activity does contribute to better survival in individuals with MetALD, other factors such as alcohol intake and its associated health risks might modulate this relationship.

The relationship between physical activity and the incidence of liver cirrhosis was less clear-cut. Kaplan–Meier curves indicated that higher physical activity was associated with a significantly lower incidence of liver cirrhosis before IPTW in the overall cohort and MASLD cohort. However, this significant association did not persist after IPTW adjustment. These results suggest that while physical activity may have a protective effect against liver cirrhosis, the effect might be influenced by confounding factors such as baseline health status, lifestyle habits, and genetic predispositions.

In the cohort without steatotic liver disease, higher physical activity was not significantly associated with liver cirrhosis incidence before or after IPTW. This lack of association might be attributed to the lower baseline risk of liver cirrhosis in this population, reducing the observable impact of physical activity.

In contrast, the MetALD cohort showed a marginally significant association between higher physical activity and lower liver cirrhosis incidence both before and after IPTW adjustment. This marginal significance presents the potential of physical activity to mitigate the risk of liver cirrhosis in individuals with concurrent metabolic dysfunction and alcohol intake, though the effect size may be modest.

Several biological mechanisms might explain the observed protective effects of physical activity on OS and liver health. Regular exercise is known to enhance hepatic lipid metabolism, reduce insulin resistance, and lower systemic inflammation—all critical factors in the pathogenesis of MASLD and liver cirrhosis [22].

Physical activity increases the oxidation of fatty acids in the liver, thereby reducing hepatic fat accumulation [23,24]. This reduction in liver fat is particularly beneficial for individuals with MASLD, as it helps to prevent the progression to more severe liver diseases such as fibrosis and cirrhosis [25,26,27].

Exercise enhances insulin sensitivity, which is crucial for individuals with metabolic dysfunction [28]. Improved insulin sensitivity helps in better glucose regulation, reducing the metabolic burden on the liver and preventing the exacerbation of liver diseases [29].

Physical activity has anti-inflammatory effects, which can help in reducing the chronic inflammation often observed in individuals with metabolic dysfunction and liver diseases [30]. Lower systemic inflammation reduces the risk of liver damage and fibrosis progression. Also, regular exercise is known to boost immune function, which can help in modulating the impact of infections and other factors that contribute to liver disease progression [31,32].

In the present study, the use of data from the UK Biobank, which includes over 500,000 participants, provides a concrete dataset for examining the associations between physical activity, OS, and liver cirrhosis. The detailed baseline data on demographic, lifestyle, and clinical characteristics, as well as liver function tests, allow for the thorough adjustment of confounding variables. Also, the application of IPTW ensures that the comparison between physical activity groups is robust and minimizes the impact of confounding factors.

While our study provides evidence for the benefits of physical activity in terms of overall survival and liver health, it is essential to offer specific recommendations for clinical practice to guide patients and healthcare providers. Activities such as brisk walking and cycling are highly recommended [33]. These exercises help improve cardiovascular health, reduce liver fat, and enhance overall metabolic function [34]. General recommendations for patients with steatotic liver disease include accumulating 150 min of moderate-intensity aerobic exercise weekly, complemented by strength and endurance training two to three times weekly. This should involve 8–10 exercises targeting major muscle groups, with 10–15 repetitions each. Patients should also aim to reduce sedentary time by incorporating short walks. Ensuring compliance with exercise routines is crucial to achieving therapeutic effects. For obese NAFLD patients, exercise programs should focus on attainable goals that lead to significant weight loss (around 10%) and improved cardiorespiratory fitness. Both aerobic and anaerobic exercises, performed for 20–60 min per session, 4–7 days weekly, for at least six months, can improve liver histology and reverse liver damage [35].

Physical activity exerts numerous beneficial effects on liver health through several biological mechanisms. These mechanisms include improvements in fatty acid metabolism, reductions in hepatic inflammation, and enhancements in liver fibrosis. Exercise, particularly structured aerobic and resistance training, activates specific receptors and pathways that contribute to these benefits. Physical activity enhances the oxidation of fatty acids in the liver, which reduces hepatic fat accumulation. This process is mediated by the activation of AMP-activated protein kinase and peroxisome proliferator-activated receptor alpha, which play critical roles in lipid metabolism and energy homeostasis [12]. Also, exercise reduces systemic inflammation and oxidative stress, which are key contributors to the progression of NAFLD. Regular physical activity decreases the levels of pro-inflammatory cytokines and increases anti-inflammatory cytokines, thereby reducing liver inflammation [36].

Nevertheless, this study has a few limitations. One of the limitations of our study is the reliance on self-reported data for physical activity frequency. Self-reported data can introduce recall bias and social desirability bias, potentially leading to inaccurate or overestimated reports of physical activity levels. This limitation is inherent in several large-scale epidemiological studies, where self-reported questionnaires are often used due to their feasibility and cost-effectiveness [37,38]. To address this potential bias, we compared our findings with those of studies that utilized objective measures of physical activity, such as accelerometers. Studies using accelerometers have consistently demonstrated a strong association between higher physical activity levels and improved health outcomes, including a reduced incidence of liver diseases and better overall survival [39,40]. These findings support the validity of our results, despite the limitations of self-reported data. Next, as an observational study, causality cannot be definitively established. While IPTW helps in approximating a randomized controlled trial, residual confounding cannot be entirely ruled out. Finally, the study does not differentiate between different types of physical activities (e.g., aerobic vs. resistance training), which may have varying effects on liver health and survival outcomes. Another limitation is the use of data from the UK Biobank, which predominantly includes participants of white ethnicity. This demographic composition may limit the generalizability of our findings to other ethnicities and populations. The prevalence and impact of steatotic liver disease can vary significantly across different demographic groups due to genetic, environmental, and lifestyle factors. Studies have shown that certain ethnic groups, such as Hispanic and Asian populations, may have higher or lower risks of developing steatotic liver disease compared to white populations [41,42]. These variations highlight the need for caution when applying our findings to more diverse populations. Also, although frequency was the primary variable used for categorization, additional data on intensity and duration were not available in the dataset. Future research should aim to include detailed measures of physical activity intensity and duration to provide a more comprehensive evaluation. Furthermore, baseline characteristics of patients with liver disease were limited to the incidence of liver cirrhosis, and detailed information on the duration of illness and medication status was not available in our dataset. Future research should investigate these variables to enable a more comprehensive analysis.

To establish causality, randomized controlled trials (RCTs) investigating the effects of different types, intensities, and frequencies of physical activity on liver health and survival outcomes are needed. RCTs could provide high-quality evidence to guide clinical recommendations. RCTs can help establish causality by minimizing confounding biases. These trials should include diverse populations to enhance generalizability. Furthermore, utilizing objective measures of physical activity, such as accelerometers, with detailed dietary and genetic data, can provide a more accurate assessment of the relationship between physical activity and liver health. In addition, future studies should incorporate objective measures of physical activity, such as accelerometers, to reduce bias and improve the accuracy of physical activity assessment. Furthermore, subsequent research is required to elucidate the precise biological mechanisms through which physical activity exerts its protective effects on liver health. Understanding these mechanisms can help in developing targeted interventions.

Long-term follow-up studies are also essential to examine the sustained impact of physical activity on liver health and overall survival. These studies can provide insights into the long-term benefits and potential risks associated with different levels of physical activity.

Given the heterogeneity in response to physical activity, future research should explore the effects of physical activity in various subpopulations, including different age groups, sexes, and ethnicities. This can help in tailoring physical activity recommendations to specific groups.

Finally, the interaction between physical activity and other lifestyle factors such as diet, alcohol consumption, and smoking should be explored. Understanding these interactions can provide a more comprehensive approach to improving liver health and survival outcomes.

## 5. Conclusions

This study provides evidence supporting the beneficial effects of regular physical activity on overall survival and its potential role in reducing the incidence of liver cirrhosis, particularly in individuals with MASLD and MetALD. The findings underscore the importance of promoting physical activity as a key strategy in public health interventions aimed at improving liver health and extending lifespan. While the protective effects of physical activity on liver cirrhosis incidence appear to be influenced by confounding factors, the consistent association with improved overall survival highlights the critical role of physical activity in enhancing overall health outcomes. Future research should focus on elucidating the mechanisms, optimizing physical activity regimens, and exploring the long-term benefits to develop comprehensive guidelines for physical activity in liver disease prevention and management.

## Figures and Tables

**Figure 1 nutrients-16-02532-f001:**
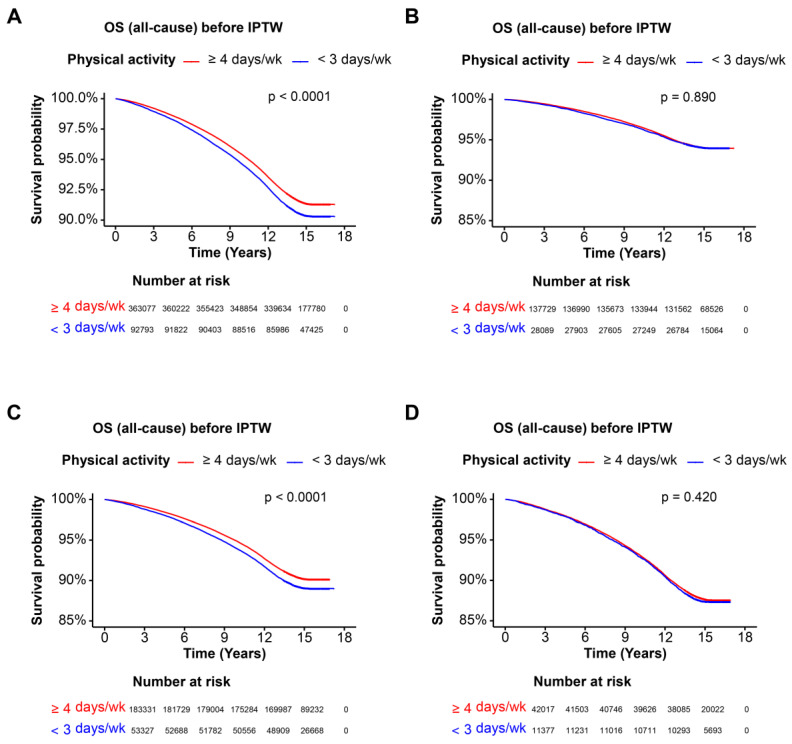
Overall survival of each cohort before IPTW. Kaplan–Meier survival curves illustrating the OS probability over time for each cohort before IPTW. Patients are categorized based on physical activity frequency: ≥4 days per week and <3 days per week. The x-axis represents the time in years, and the y-axis represents the survival probability. The log-rank test *p*-value indicates the statistical significance of the survival difference between the groups. (**A**) OS of entire cohort, (**B**) OS of cohort without steatotic liver disease, (**C**) OS of MASLD cohort, (**D**) OS of MetALD cohort. *p* < 0.05 as significant result. Abbreviations: OS, overall survival; IPTW, inverse probability of treatment weighting; MASLD, metabolic dysfunction-associated steatotic liver disease; MetALD, MASLD and increased alcohol intake.

**Figure 2 nutrients-16-02532-f002:**
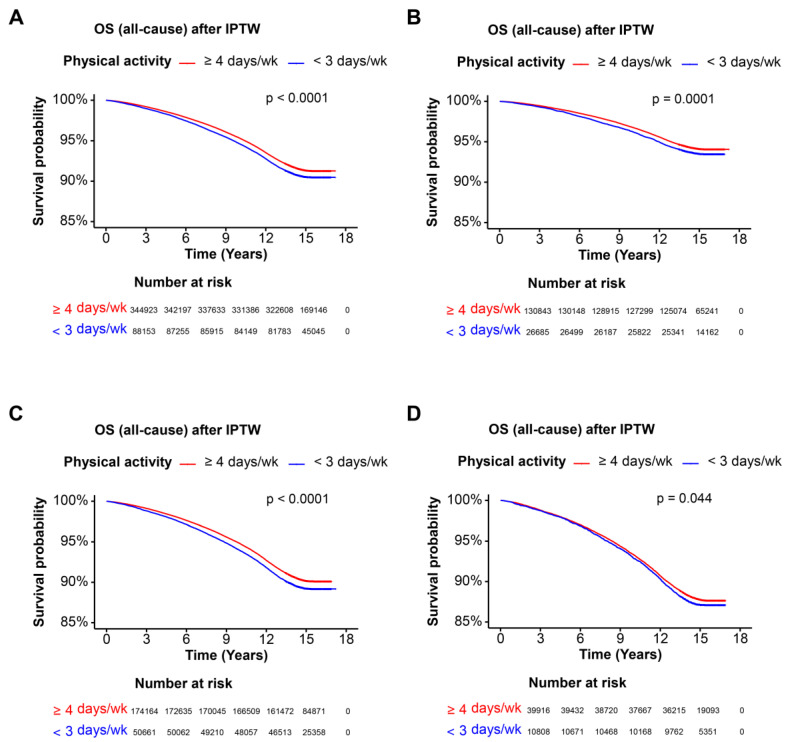
Overall survival of each cohort after IPTW. Kaplan–Meier survival curves illustrating the OS probability over time for each cohort after IPTW. Patients are categorized based on physical activity frequency: ≥4 days per week and <3 days per week. The x-axis represents the time in years, and the y-axis represents the survival probability. The log-rank test *p*-value indicates the statistical significance of the survival difference between the groups. (**A**) OS of entire cohort, (**B**) OS of cohort without steatotic liver disease, (**C**) OS of MASLD cohort, (**D**) OS of MetALD cohort. *p* < 0.05 as significant result. Abbreviations: OS, overall survival; IPTW, inverse probability of treatment weighting; MASLD, metabolic dysfunction-associated steatotic liver disease; MetALD, MASLD and increased alcohol intake.

**Figure 3 nutrients-16-02532-f003:**
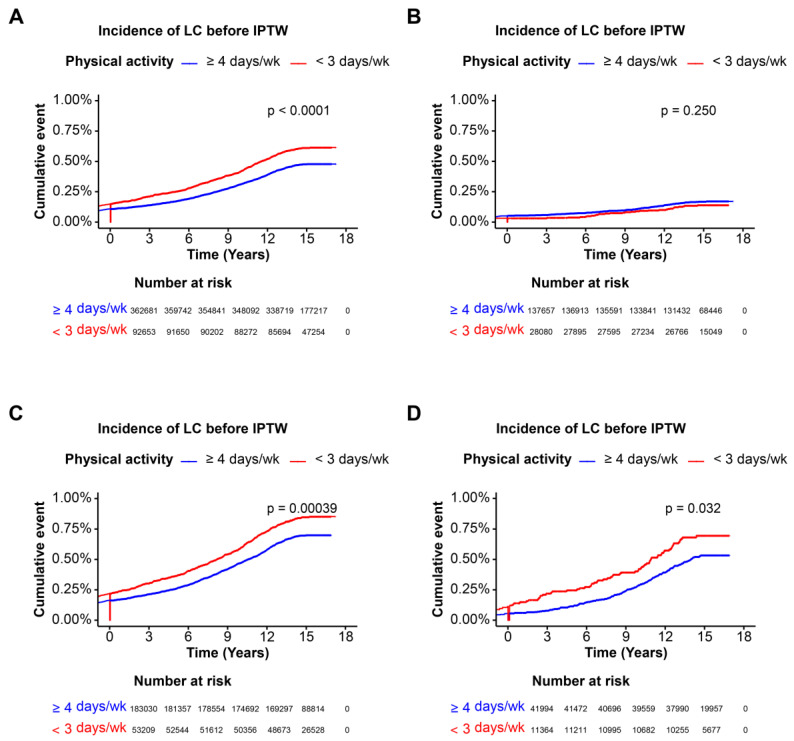
Incidence of liver cirrhosis in each cohort before IPTW. Kaplan–Meier curves illustrating liver cirrhosis incidence over time for participants of each cohort, categorized by physical activity frequency, before IPTW. The log-rank test *p*-value indicates the significance of incidence differences. (**A**) OS of entire cohort, (**B**) OS of cohort without steatotic liver disease, (**C**) OS of MASLD cohort, (**D**) OS of MetALD cohort. *p* < 0.05 as significant result. Abbreviations: OS, overall survival; IPTW, inverse probability of treatment weighting; MASLD, metabolic dysfunction-associated steatotic liver disease; MetALD, MASLD and increased alcohol intake.

**Figure 4 nutrients-16-02532-f004:**
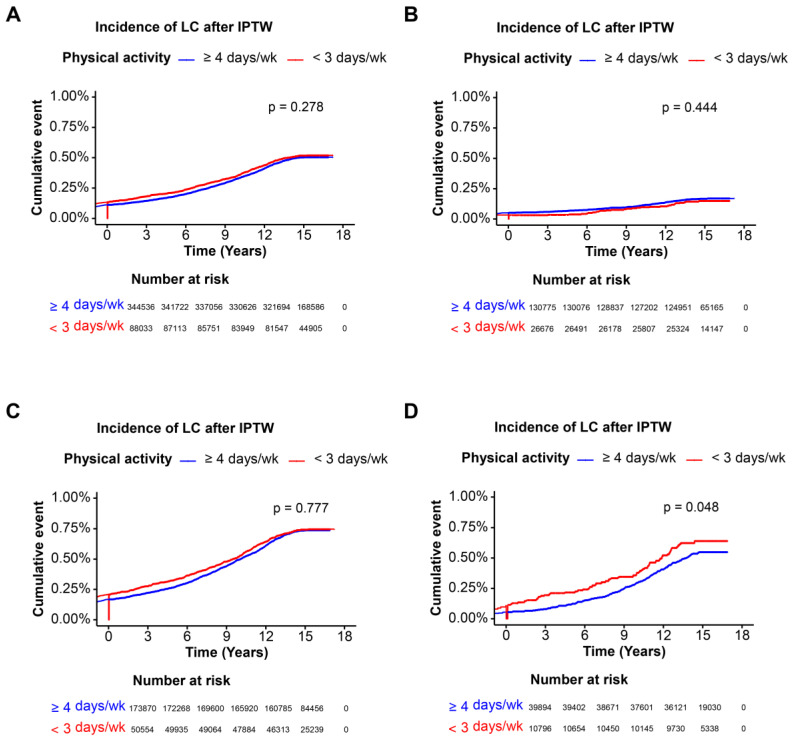
Incidence of liver cirrhosis in each cohort after IPTW. Kaplan–Meier curves illustrating liver cirrhosis incidence over time for participants of each cohort, categorized by physical activity frequency, after IPTW. The log-rank test *p*-value indicates the significance of incidence differences. (**A**) OS of entire cohort, (**B**) OS of cohort without steatotic liver disease, (**C**) OS of MASLD cohort, (**D**) OS of MetALD cohort. *p* < 0.05 as significant result. Abbreviations: OS, overall survival; IPTW, inverse probability of treatment weighting; MASLD, metabolic dysfunction-associated steatotic liver disease; MetALD, MASLD and increased alcohol intake.

**Table 1 nutrients-16-02532-t001:** Baseline clinical characteristics of the entire cohort.

Clinical Characteristics	Before IPTW			After IPTW		
	Physical Activity		SMD	Physical Activity		SMD
	≥4 Days/Week, N = 363,077	<3 Days/Week, N = 92,793		≥4 Days/Week, N = 344,923	<3 Days/Week, N = 88,153	
Sex (Male)	160,651 (44.2%)	44,676 (48.1%)	0.079	155,386 (45.0%)	39,736 (45.1%)	0.001
Age at recruitment	56.74 ± 8.12	55.80 ± 8.00	0.116	56.54 ± 8.16	56.46 ± 7.92	0.011
Smoking status			0.023			0.002
Current	37,089 (10.2%)	10,115 (10.9%)		35,734 (10.4%)	9168 (10.4%)	
Previous	125,037 (34.4%)	31,565 (34.1%)		118,477 (34.3%)	30,176 (34.2%)	
Never	200,951 (55.3%)	50,992 (55.0%)		190,713 (55.3%)	48,786 (55.4%)	
Body mass index (kg/m^2^)	27.19 ± 4.63	28.25 ± 5.32	0.213	27.40 ± 4.76	27.39 ± 4.90	0.002
Waist circumference (cm)	89.48 ± 13.19	92.69 ± 14.22	0.234	90.13 ± 13.43	90.11 ± 13.59	0.002
Type 2 diabetes	35,740 (9.8%)	11,812 (12.7%)	0.091	35,972 (10.4%)	9170 (10.4%)	0.001
Dyslipidemia	102,198 (28.1%)	27,792 (30.0%)	0.040	98,314 (28.5%)	24,991 (28.3%)	0.003
Hypertension	148,883 (41.0%)	39,998 (43.1%)	0.043	142,900 (41.4%)	36,446 (41.3%)	0.002
ALT (U/L)	23.06 ± 13.54	24.45 ± 14.57	0.099	23.35 ± 13.89	23.35 ± 13.53	<0.001
GGT (U/L)	35.79 ± 38.26	39.14 ± 42.44	0.083	36.49 ± 39.96	36.57 ± 37.73	0.002
Platelet (10^9^/L)	253.08 ± 59.71	254.69 ± 60.73	0.027	253.41 ± 59.96	253.46 ± 59.94	0.002
Albumin (g/L)	4.52 ± 0.26	4.51 ± 0.27	0.036	4.52 ± 0.26	4.52 ± 0.27	0.001

Data are described as mean ± standard deviation or n (%). ALT, alanine transaminase; GGT, gamma-glutamyl transferase; SMD, standardized mean difference.

**Table 2 nutrients-16-02532-t002:** Baseline clinical characteristics of the participants without steatotic liver disease.

Clinical Characteristics	Before IPTW			After IPTW		
	Physical Activity		SMD	Physical Activity		SMD
	≥4 Days/Week, N = 137,729	<3 Days/Week, N = 28,089		≥4 Days/Week, N = 130,843	<3 Days/Week, N = 26,685	
Sex (Male)	30,847 (22.4%)	7067 (25.2%)	0.065	29,910 (22.9%)	6079 (22.8%)	0.002
Age at recruitment	55.37 ± 8.27	54.02 ± 8.03	0.166	55.14 ± 8.29	55.08 ± 8.00	0.007
Smoking status			0.039			0.003
Current	12,996 (9.4%)	2669 (9.5%)		12,375 (9.5%)	2543 (9.5%)	
Previous	40,643 (29.5%)	7759 (27.7%)		38,225 (29.2%)	7805 (29.3%)	
Never	84,090 (61.1%)	17,540 (62.7%)		80,243 (61.3%)	16,317 (61.2%)	
Body mass index (kg/m^2^)	23.52 ± 2.34	23.57 ± 2.36	0.021	23.53 ± 2.34	23.53 ± 2.37	0.003
Waist circumference (cm)	77.51 ± 7.11	78.08 ± 7.09	0.080	77.61 ± 7.12	77.61 ± 7.05	<0.001
Type 2 diabetes	4005 (2.9%)	825 (2.9%)	0.002	3804 (2.9%)	779 (2.9%)	0.001
Dyslipidemia	21,687 (15.7%)	4047 (14.4%)	0.037	20,304 (15.5%)	4138 (15.5%)	<0.001
Hypertension	34,452 (25.0%)	6608 (23.5%)	0.035	32,399 (24.8%)	6603 (24.7%)	<0.001
ALT (U/L)	17.48 ± 7.68	17.31 ± 7.80	0.022	17.45 ± 7.61	17.47 ± 8.36	0.003
GGT (U/L)	21.62 ± 13.10	21.51 ± 13.11	0.009	21.60 ± 13.11	21.59 ± 13.00	0.001
Platelet (10^9^/L)	17.48 ± 7.68	17.31 ± 7.80	0.023	254.19 ± 59.01	254.28 ± 58.12	0.002
Albumin (g/L)	4.53 ± 0.26	4.53 ± 0.26	0.009	4.53 ± 0.26	4.53 ± 0.26	0.001

Data are described as mean ± standard deviation or n (%). ALT, alanine transaminase; GGT, gamma-glutamyl transferase; SMD, standardized mean difference.

**Table 3 nutrients-16-02532-t003:** Baseline clinical characteristics of the patients with metabolic dysfunction-associated steatotic liver disease (MASLD).

Clinical Characteristics	Before IPTW			After IPTW		
	Physical Activity		SMD	Physical Activity		SMD
	≥4 Days/Week, N = 183,331	<3 Days/Week, N = 53,327		≥4 Days/Week, N = 174,164	<3 Days/Week, N = 50,661	
Sex (Male)	99,985 (54.5%)	29,588 (55.5%)	0.019	95,356 (54.8%)	27,720 (54.7%)	0.001
Age at recruitment	57.23 ± 8.01	56.29 ± 7.93	0.118	57.01 ± 8.05	56.93 ± 7.84	0.010
Smoking status			0.031			0.002
Current	18,479 (10.1%)	5890 (11.0%)		17,941 (10.3%)	5240 (10.3%)	
Previous	63,951 (34.9%)	18,352 (34.4%)		60,547 (34.8%)	17,573 (34.7%)	
Never	100,901 (55.0%)	29,085 (54.5%)		95,677 (54.9%)	27,848 (55.0%)	
Body mass index (kg/m^2^)	29.68 ± 4.34	30.53 ± 5.08	0.181	29.87 ± 4.48	29.87 ± 4.65	0.001
Waist circumference (cm)	96.87 ± 10.59	99.15 ± 11.84	0.203	97.38 ± 10.89	97.37 ± 11.04	0.001
Type 2 diabetes	27,371 (14.9%)	9564 (17.9%)	0.081	27,180 (15.6%)	7899 (15.6%)	<0.001
Dyslipidemia	64,719 (35.3%)	19,575 (36.7%)	0.029	62,008 (35.6%)	17,974 (35.5%)	0.003
Hypertension	92,103 (50.2%)	27,395 (51.4%)	0.023	87,924 (50.5%)	25,534 (50.4%)	0.002
ALT (U/L)	26.24 ± 14.89	27.34 ± 15.68	0.072	26.50 ± 15.32	26.54 ± 14.67	0.003
GGT (U/L)	41.99 ± 41.48	44.55 ± 44.24	0.060	42.60 ± 43.41	42.77 ± 40.07	0.004
Platelet (10^9^/L)	254.01 ± 60.80	255.76 ± 62.17	0.028	254.41 ± 61.06	254.44 ± 61.48	0.001
Albumin (g/L)	4.51 ± 0.26	4.50 ± 0.27	0.044	4.51 ± 0.26	4.51 ± 0.27	0.003

Data are described as mean ± standard deviation or n (%). ALT, alanine transaminase; GGT, gamma-glutamyl transferase; SMD, standardized mean difference.

**Table 4 nutrients-16-02532-t004:** Baseline clinical characteristics of the patients with MetALD.

Clinical Characteristics	Before IPTW			After IPTW		
	Physical Activity		SMD	Physical Activity		SMD
	≥4 Days/Week, N = 42,017	<3 Days/Week, N = 11,377		≥4 Days/Week, N = 39,916	<3 Days/Week, N = 10,808	
Sex (Male)	29,819 (71.0%)	8021 (70.5%)	0.010	28,287 (70.9%)	7657 (70.8%)	<0.001
Age at recruitment	59.13 ± 7.33	57.92 ± 7.40	0.165	58.87 ± 7.43	58.78 ± 7.15	0.012
Smoking status			0.015			0.006
Current	5614 (13.4%)	1556 (13.7%)		5366 (13.4%)	1472 (13.6%)	
Previous	20,443 (48.7%)	5454 (47.9%)		19,352 (48.5%)	5220 (48.3%)	
Never	15,960 (38.0%)	4367 (38.4%)		15,198 (38.1%)	4116 (38.1%)	
Body mass index (kg/m^2^)	28.35 ± 3.53	29.07 ± 4.06	0.189	28.50 ± 3.64	28.49 ± 3.74	0.003
Waist circumference (cm)	96.48 ± 9.60	98.32 ± 10.70	0.180	96.87 ± 9.82	96.84 ± 10.06	0.001
Type 2 diabetes	4364 (10.4%)	1423 (12.5%)	0.067	4323 (10.8%)	1163 (10.8%)	0.002
Dyslipidemia	15,792 (37.6%)	4170 (36.7%)	0.019	14,914 (37.4%)	4019 (37.2%)	0.004
Hypertension	22,328 (53.1%)	5995 (52.7%)	0.009	21,168 (53.0%)	5720 (52.9%)	0.002
ALT (U/L)	27.49 ± 16.08	28.47 ± 15.72	0.062	27.70 ± 16.29	27.70 ± 14.99	<0.001
GGT (U/L)	55.17 ± 58.15	57.18 ± 62.05	0.034	55.60 ± 58.91	55.60 ± 59.33	<0.001
Platelet (10^9^/L)	246.15 ± 57.31	248.17 ± 58.60	0.035	246.58 ± 57.49	246.71 ± 57.96	0.002
Albumin (g/L)	4.54 ± 0.26	4.53 ± 0.26	0.029	4.54 ± 0.26	4.54 ± 0.26	0.002

Data are described as mean ± standard deviation or n (%). ALT, alanine transaminase; GGT, gamma-glutamyl transferase; SMD, standardized mean difference.

**Table 5 nutrients-16-02532-t005:** Impact of less physical activity on all-cause mortality.

**Cohorts**	**Before IPTW**			
	**Univariate Analysis**		**Multivariate Analysis**	
	**HR (95% CI)**	***p* Value**	**HR (95% CI)**	***p* Value**
Entire cohort	1.12 (1.10, 1.15)	<0.001	1.12 (1.10, 1.15)	<0.001
Participants without SLD	1.00 (0.95, 1.06)	0.890	-	-
MASLD	1.13 (1.09, 1.16)	<0.001	1.12 (1.08, 1.15)	0.001
MetALD	1.02 (0.97, 1.02)	0.420	-	-
**Cohorts**	**After IPTW**			
	**Univariate Analysis**		**Multivariate Analysis**	
	**HR (95% CI)**	***p* Value**	**HR (95% CI)**	***p* Value**
Entire cohort	1.10 (1.07, 1.13)	<0.001	1.12 (1.10, 1.15)	<0.001
Participants without SLD	1.11 (1.05, 1.17)	<0.001	1.16 (1.10, 1.23)	<0.001
MASLD	1.11 (1.07, 1.14)	<0.001	1.12 (1.09, 1.16)	<0.001
MetALD	1.06 (0.99, 1.12)	0.044	1.07 (1.00, 1.14)	0.039

HR, hazard ratio; CI, confidence interval; SLD, steatotic liver disease; MASLD, metabolic dysfunction-associated steatotic liver disease.

## Data Availability

The original contributions presented in the study are included in the article, further inquiries can be directed to the corresponding authors.

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
