# Peer review of "Impact of Physical Activity on Overall Survival and Liver Cirrhosis Incidence in Steatotic Liver Disease: Insights from a Large Cohort Study Using Inverse Probability of Treatment Weighting"

_nutrients, 2024, doi:10.3390/nu16152532_

Round 1

Reviewer 1 Report

Comments and Suggestions for Authors

This study investigates the relationship between physical activity frequency, overall survival, and the incidence of liver cirrhosis in a large cohort with or without steatotic liver diseases. The conclusion showed that promoting physical activity may be key in improving liver health and survival outcomes. Here is some comments.
1. The abstract provides a clear overview of the study's purpose and findings. However, it could be more cohesive by explicitly stating the primary outcomes of interest before delving into the methodology. Consider reordering the abstract to first present the study's main objectives, followed by the methods and key results.
2. The use of Inverse Probability of Treatment Weighting (IPTW) is appropriate for balancing baseline characteristics. However, the manuscript would benefit from a more detailed description of the IPTW process, including the selection of covariates and the rationale behind their choice. Additionally, a sensitivity analysis to assess the robustness of the IPTW results would strengthen the conclusions.
3. The manuscript relies on self-reported data for physical activity frequency, which may introduce bias. It is recommended that the authors discuss this potential limitation and possibly compare their findings with studies that have used objective measures of physical activity, such as accelerometers.
4. The study utilizes data from the UK Biobank, which predominantly includes participants of white ethnicity. The authors should discuss the generalizability of their findings to other ethnicities and populations, as the prevalence and impact of steatotic liver disease may vary across different demographic groups.
5. While the study provides evidence for the benefits of physical activity on overall survival and liver health, it lacks specific recommendations for clinical practice. The authors could consider providing evidence-based suggestions for the type, intensity, and duration of physical activity that could be beneficial for patients with steatotic liver disease.
6.The discussion section touches on biological mechanisms that might explain the protective effects of physical activity on liver health. It would be beneficial to delve deeper into these mechanisms, possibly including a schematic representation to aid understanding.
7. The authors have acknowledged the limitations of their study, including reliance on self-reported data and the inability to establish causality. It is suggested that they also discuss potential residual confounding factors and how future research might address these issues, including the use of randomized controlled trials.
8. The Kaplan-Meier curves and other figures are essential for visualizing the study's findings. However, the quality of these figures in the provided manuscript is not assessable. The authors should ensure that all figures are clear, well-labeled, and of high resolution in the final submission.
9. The manuscript is well-written, but a thorough proofreading pass for grammar, punctuation, and typographical errors is recommended to ensure the highest standard of English language quality.

Comments on the Quality of English Language

Moderate editing of English language required.

Author Response

This study investigates the relationship between physical activity frequency, overall survival, and the incidence of liver cirrhosis in a large cohort with or without steatotic liver diseases. The conclusion showed that promoting physical activity may be key in improving liver health and survival outcomes. Here is some comments.

  1. The abstract provides a clear overview of the study's purpose and findings. However, it could be more cohesive by explicitly stating the primary outcomes of interest before delving into the methodology. Consider reordering the abstract to first present the study's main objectives, followed by the methods and key results.

A: Thank you for your insightful comments and suggestions. We have revised the abstract to explicitly state the primary and secondary outcomes of interest before describing the methodology.

  1. The use of Inverse Probability of Treatment Weighting (IPTW) is appropriate for balancing baseline characteristics. However, the manuscript would benefit from a more detailed description of the IPTW process, including the selection of covariates and the rationale behind their choice. Additionally, a sensitivity analysis to assess the robustness of the IPTW results would strengthen the conclusions.

A: Thank you for your valuable comment. We appreciate your suggestions to enhance the manuscript. We have provided a more detailed description of the IPTW process, including the selection of covariates and the rationale behind their choice.

Regarding the sensitivity analysis to assess the robustness of the IPTW results, we have carefully considered your recommendation. Due to the nature of our study and the characteristics of the variables involved, conducting a comprehensive sensitivity analysis is challenging. However, we have performed additional checks to ensure the robustness of our findings.

  1. The manuscript relies on self-reported data for physical activity frequency, which may introduce bias. It is recommended that the authors discuss this potential limitation and possibly compare their findings with studies that have used objective measures of physical activity, such as accelerometers.

A: Thank you for your valuable feedback. We appreciate your suggestion to discuss the potential limitation introduced by relying on self-reported data for physical activity frequency and to compare our findings with studies that have used objective measures of physical activity. We have addressed in the revised manuscript.

  1. The study utilizes data from the UK Biobank, which predominantly includes participants of white ethnicity. The authors should discuss the generalizability of their findings to other ethnicities and populations, as the prevalence and impact of steatotic liver disease may vary across different demographic groups.

A: We appreciate your feedback and for highlighting the importance of discussing the generalizability of our findings. We appreciate your suggestion and have addressed this point in the revised manuscript.

  1. While the study provides evidence for the benefits of physical activity on overall survival and liver health, it lacks specific recommendations for clinical practice. The authors could consider providing evidence-based suggestions for the type, intensity, and duration of physical activity that could be beneficial for patients with steatotic liver disease.

A: Thank you for your important comment. We appreciate your suggestion to provide evidence-based recommendations for clinical practice regarding the type, intensity, and duration of physical activity beneficial for patients with steatotic liver disease. We have addressed this point in the revised manuscript by including specific recommendations supported by recent research.

  1. The discussion section touches on biological mechanisms that might explain the protective effects of physical activity on liver health. It would be beneficial to delve deeper into these mechanisms, possibly including a schematic representation to aid understanding.

A: We appreciate for the comment. We agree to your suggestion to delve deeper into the biological mechanisms that explain the protective effects of physical activity on liver health and to possibly include a schematic representation to aid understanding. We have written this review point in the revised manuscript by expanding on the biological mechanisms and providing references to support these details.

  1. The authors have acknowledged the limitations of their study, including reliance on self-reported data and the inability to establish causality. It is suggested that they also discuss potential residual confounding factors and how future research might address these issues, including the use of randomized controlled trials.

A: Thank you for your comment. We appreciate your suggestion to discuss potential residual confounding factors and how future research might address these issues, including the use of randomized controlled trials. We have addressed this point in the revised manuscript.

  1. The Kaplan-Meier curves and other figures are essential for visualizing the study's findings. However, the quality of these figures in the provided manuscript is not assessable. The authors should ensure that all figures are clear, well-labeled, and of high resolution in the final submission.

A: Thank you for the comment. We will ensure that all figures in the final submission are of high resolution, clearly labeled, and properly formatted to enhance readability and visual impact. We recognize the critical role these figures play in illustrating our study's results and will take the necessary steps to improve their quality.

  1. The manuscript is well-written, but a thorough proofreading pass for grammar, punctuation, and typographical errors is recommended to ensure the highest standard of English language quality.

A: Thank you for the feedback. We have thoroughly proofread the manuscript to address grammar, punctuation, and typographical errors to ensure the highest standard of English language quality. We appreciate your recommendation, and we believe the revised manuscript now meets the necessary standards.

Reviewer 2 Report

Comments and Suggestions for Authors

This study was aimed to investigate the relationship between physical activity frequency, overall survival, and the incidence of liver cirrhosis in a large cohort with or without steatotic liver diseases, including metabolic dysfunction associated steatotic liver disease (MASLD) and MASLD and increased alcohol intake (MetALD). As the results, this study demonstrates that higher physical activity was associated with significantly better OS in the overall cohort and MASLD cohort. Moreover, in participants without steatotic liver disease and the MetALD cohort, higher physical showed significant OS improvement. The reviewer considers the results of this study to be useful clinical information from the perspective of preventive liver disease and survival prognosis. However, there are several limitations in this study.

1.     As the authors explain in the “Introduction (L57-58)” and “Study limitation (L355-357)” sections, physical activity is defined by the type, intensity, duration, and frequency of exercise. As the authors are aware, physical activity is evaluated as the product of exercise intensity, duration, and frequency. However, this study only categorized physical activity into ≥4 days per week and <4 days per week, and the authors only assessed frequency of exercise. The reviewer believes that the results of this study did not rigorously evaluate physical activity.

2.     This study targets patients with liver disease, and also includes the onset of liver disease as a study outcome. Although this study was a liver study, it lacked information on alcohol intake. Especially, the reviewer thinks that information on alcohol intake is essential for the evaluation of MASLD patients with a large alcohol consumption (MetALD).

Additionally, how did the authors handle the data on alcohol intake when conducting their statistical analysis? Please explain how the authors handled the data on alcohol intake in the statistical analysis of this study.

3.     This study also had little baseline patient information on patients with liver disease (MASLD, MetALD, and liver cirrhosis). The reviewer thinks that information such as the type and severity of liver disease, duration of illness, and medication status is necessary.

4.     In relation to the above, the reviewer considers that γ-GTP data are also necessary when conducting liver disease studies, especially in patients with MetALD. Authors should also provide data for γ-GTP if they have it.

5.     This study did not provide definitions of the observation period or participant endpoints. From the figure, it appears that the patients were followed up for up to 18 years, but the authors should explain the observation period and the participants' endpoints in the text.

6.     In this study, the authors used inverse probability of treatment weighting (IPTW) as the statistical analysis method. However, there is little description of IPTW in the “Methods” section. The authors should provide a detailed explanation of IPTW so that non-experts can understand this method.

7.     In this study, the authors did not provide a definition of liver disease (MASLD, MetALD, and liver cirrhosis). The authors should provide definitions of liver disease (MASLD, MetALD, and liver cirrhosis).

Author Response

This study was aimed to investigate the relationship between physical activity frequency, overall survival, and the incidence of liver cirrhosis in a large cohort with or without steatotic liver diseases, including metabolic dysfunction associated steatotic liver disease (MASLD) and MASLD and increased alcohol intake (MetALD). As the results, this study demonstrates that higher physical activity was associated with significantly better OS in the overall cohort and MASLD cohort. Moreover, in participants without steatotic liver disease and the MetALD cohort, higher physical showed significant OS improvement. The reviewer considers the results of this study to be useful clinical information from the perspective of preventive liver disease and survival prognosis. However, there are several limitations in this study.

  1. As the authors explain in the “Introduction (L57-58)” and “Study limitation (L355-357)” sections, physical activity is defined by the type, intensity, duration, and frequency of exercise. As the authors are aware, physical activity is evaluated as the product of exercise intensity, duration, and frequency. However, this study only categorized physical activity into ≥4 days per week and <4 days per week, and the authors only assessed frequency of exercise. The reviewer believes that the results of this study did not rigorously evaluate physical activity.

A: Thank you for your thorough and insightful feedback on our manuscript. We appreciate your constructive comments and suggestions, which will help improve the clarity and rigor of our study.

We acknowledge the limitation of using only frequency to categorize physical activity. Unfortunately, due to limitations in the dataset, we were unable to evaluate the intensity and duration of physical activity. We recognize the importance of these factors and suggest that future studies should aim to collect comprehensive physical activity data to enable a more rigorous evaluation. We addressed these points for the limitation of study.

  1. This study targets patients with liver disease, and also includes the onset of liver disease as a study outcome. Although this study was a liver study, it lacked information on alcohol intake. Especially, the reviewer thinks that information on alcohol intake is essential for the evaluation of MASLD patients with a large alcohol consumption (MetALD).

Additionally, how did the authors handle the data on alcohol intake when conducting their statistical analysis? Please explain how the authors handled the data on alcohol intake in the statistical analysis of this study.

A: We agree with the reviewer and have ensured that our analysis and categorization of MASLD and MetALD are aligned with established guidelines. Alcohol intake data were collected through self-reported questionnaires, and for statistical analysis, alcohol consumption was categorized according to established thresholds (20-50 g/day for females and 30-60 g/day for males) to identify MetALD patients. This approach follows the guidelines for defining and differentiating MASLD and MetALD. We addressed the regarding contents in the method section

  1. This study also had little baseline patient information on patients with liver disease (MASLD, MetALD, and liver cirrhosis). The reviewer thinks that information such as the type and severity of liver disease, duration of illness, and medication status is necessary.

A: We appreciate the reviewer's suggestion. Due to limitations in our dataset, we were only able to obtain information on the incidence of liver cirrhosis (LC). Unfortunately, detailed data on the duration of illness and medication status were not available. We acknowledge the importance of these factors and recommend that future studies collect comprehensive baseline data to provide a more detailed analysis. Additional contents were addressed in discussion section.

  1. In relation to the above, the reviewer considers that γ-GTP data are also necessary when conducting liver disease studies, especially in patients with MetALD. Authors should also provide data for γ-GTP if they have it.

A: We agree that γ-GTP is an important marker, particularly in the context of alcohol-related liver diseases. We have included γ-GTP data in the baseline characteristics and incorporated it into our analysis for IPTW. Recognizing its significance, we acknowledge that further analysis of γ-GTP levels could provide valuable insights into MASLD and MetALD.

  1. This study did not provide definitions of the observation period or participant endpoints. From the figure, it appears that the patients were followed up for up to 18 years, but the authors should explain the observation period and the participants' endpoints in the text.

A: We appreciate the opportunity to clarify these details. The observation period for this study spanned up to 18 years, with an average follow-up period of approximately 15 years. Participants were followed from the baseline assessment until the occurrence of the endpoint, loss to follow-up, or the end of the study period, and we added the information in the manuscript. The primary outcome was overall survival (OS), and secondary outcomes included the incidence of liver cirrhosis and other liver-related outcomes. These outcomes were identified through hospital records and death registries. We have used the terms "primary outcome" and "secondary outcomes" interchangeably with "primary endpoint" and "secondary endpoints" throughout the manuscript for consistency.

  1. In this study, the authors used inverse probability of treatment weighting (IPTW) as the statistical analysis method. However, there is little description of IPTW in the “Methods” section. The authors should provide a detailed explanation of IPTW so that non-experts can understand this method.

A: We appreciate the importance of providing a detailed explanation of IPTW for non-experts. Therefore, we have expanded the Methods section to include a comprehensive description of the IPTW process.

  1. In this study, the authors did not provide a definition of liver disease (MASLD, MetALD, and liver cirrhosis). The authors should provide definitions of liver disease (MASLD, MetALD, and liver cirrhosis).

A: We appreciate for your comment to clarify these definitions. We have included the definitions of MASLD, MetALD, and liver cirrhosis in the revised manuscript as derived from established sources including the recent studies.

Round 2

Reviewer 2 Report

Comments and Suggestions for Authors

I think all responses to reviewers' comments have been addressed satisfactorily.

I have no comments on the revised manuscript.